# The common house spider, *Parasteatoda tepidariorum*, maintains silk gene expression on sub-optimal diet

Jeremy Miller¤, Jannelle Vienneau-Hathaway, Enkhbileg Dendev, Merrina Lan, Nadia A. Ayoub *

Department of Biology, Washington and Lee University, Lexington, VA, United States of America

¤ Current address: University of Kentucky College of Medicine, Lexington, KY, United States of America
* ayoubn@wlu.edu

**Data Availability Statement:** Newly sequenced cDNA clones have been deposited in GenBank (MH367500, MH367501). All other relevant data

## Abstract

Cobweb weaving spiders and their relatives spin multiple task-specific fiber types. The unique material properties of each silk type result from differences in amino acid sequence and structure of their component proteins, primarily spidroins (spider fibrous proteins). Amino acid content and gene expression measurements of spider silks suggest some spiders change expression patterns of individual protein components in response to environmental cues. We quantified mRNA abundance of three spidroin encoding genes involved in prey capture in the common house spider, *Parasteatoda tepidariorum* (Theridiidae), fed different diets. After 10 days of acclimation to the lab on a diet of mealworms, spiders were split into three groups: (1) individuals were immediately dissected, (2) spiders were fed high-energy crickets, or (3) spiders were fed low-energy flies, for 1 month. All spiders gained mass during the acclimation period and cricket-fed spiders continued to gain mass, while fly-fed spiders either maintained or lost mass. Using quantitative PCR, we found no significant differences in the absolute or relative abundance of dragline gene transcripts, major ampullate spidroin 1 (*MaSp1*) and major ampullate spidroin 2 (*MaSp2*), among groups. In contrast, prey-wrapping minor ampullate spidroin (*MiSp*) gene transcripts were significantly less abundant in fly-fed than lab-acclimated spiders. However, when measured relative to *Actin*, cricket-fed spiders showed the lowest expression of *MiSp*. Our results suggest that house spiders are able to maintain silk production, even in the face of a low-quality diet.

## Introduction

Spiders synthesize silk in specialized abdominal glands, and most have multiple morphologically and functionally differentiated gland types. Orb-web and cobweb weaving spiders and their relatives (superfamily Araneoidea) have seven such glands [1]. For instance, silk spun from the major ampullate glands is used both as dragline silk and structural silk to build the frame and radii of the orb-web. Orb-web weaving spiders then spin an auxiliary spiral using silk made in minor ampullate glands, which keeps the body of the web stabilized until it is replaced by the permanent capture spiral threads [2]. Cobweb weaving araneoid spiders

are within the manuscript or its Supporting Information files.

**Funding:** Funding for this work was provided by the Howard Hughes Medical Institute (52007570) to support the salary of JM, the National Science Foundation, Biology Directorate (1755142) to NAA, and Washington and Lee Unviersity through Summer Lenfest Research Support to NAA, and through the Summer Research Scholars Program to support the salaries of JV-H, ED, and ML. The funders had no role in study design, data collection and analysis, decision to publish, or preparation of the manuscript.

**Competing interests:** The authors have declared that no competing interests exist.

(Theridiidae) also use major ampullate silk as draglines and for the majority of the web [3]. Cobweb weavers do not spin a capture spiral but use minor ampullate silk for prey-wrapping [4].

Spidroins (spider fibrous proteins) are encoded by a gene family found only in spiders [5]. The specialized functions and unique material properties of different spider silk types are the product of composite spidroins [5–7]. For instance, major ampullate glands express major ampullate spidroin 1 (MaSp1) and major ampullate spidroin 2 (MaSp2) [8, 9]. Both proteins have a high proportion of β-sheet forming poly-alanine stretches that likely confer strength. However, MaSp2 contains numerous glycine-proline-glycine (GPG) motifs which form β-turn spirals [6, 7, 10–12]. Proline reduces protein alignment when in high abundance in major ampullate silk [13], and may also be hydroxylated after translation [14, 15], both of which should contribute to the elasticity of the major ampullate silk. Indeed, across 85 species, those with a higher percentage of the proline-containing MaSp2 relative to the proline-poor MaSp1 have more extensible silk [13, 15]. Therefore, the ratio of MaSp1 to MaSp2 is an important determinant of silk mechanical properties, namely extensibility and strength. Minor ampullate silks, composed of minor ampullate spidroins (MiSp), are typically not as strong but are more extensible than major ampullate silks [16, 17]. Although most published MiSp sequences lack proline, the MiSp found in a particular cobweb weaver, the false black widow *Steatoda grossa*, has a high proportion of GPG motifs which is associated with this species having the most extensible minor ampullate silks [18].

Relative spidroin composition and spider silk mechanical properties can respond to environmental conditions (reviewed in [19]). Changes in amino acid content of dragline silks serve as support for changes in spidroin expression levels in response to diet [20–24]. Protein-rich diets lead to greater fiber strength and extensibility [24]. Under protein deprivation, spider silk is lower in percentage of glutamine, glycine, and proline. Because glutamine and proline are abundant in MaSp2 and extremely limited in MaSp1, it was inferred that MaSp1 is preferentially expressed over MaSp2 under protein constraints [24]. Conflicting results have been found for spidroin expression of the orb-weaver *Nephila pilipes* fed flies versus crickets. Tso et al. [20] found that fly-fed spiders produced major ampullate silk with a higher percentage of alanine and lower percentages of glutamine and proline than cricket fed spiders, suggesting higher MaSp1:MaSp2 ratio in fly-fed spiders. In contrast, Blamires et al. [22] inferred higher MaSp1:MaSp2 in cricket-fed spiders as they found the major ampullate silks of fly-fed spiders were lower in glutamine, glycine, and alanine than cricket-fed spiders. Since the proportion of amino acids cannot be measured independently of each other, and the sequences of MaSp1 and MaSp2 were unknown for the species in these experimental studies, amino acid composition may be a poor indicator of absolute or even relative spidroin expression.

Blamires et al. (2018) [25] tested the effect of protein deprivation on multiple properties of major ampullate silk for five species of araneoid spiders, including amino acid content of major ampullate silk and expression levels of *MaSp1* and *MaSp2* in major ampullate glands. In this study, all species showed a significant difference in *MaSp1* or *MaSp2* expression levels between protein-deprived and protein-fed spiders, but the direction of change varied for each species. Additionally, three of the species showed significant differences in amino acid content as a consequence of protein deprivation. However, only one species' change in amino acid content reflected the change in gene expression levels; *Phonognatha graefei* increased proline in protein-deprived spiders, which is consistent with its observed downregulation of *MaSp1* and upregulation of *MaSp2* under protein-deprivation. In contrast, protein deprived *Argiope keyserlingi* upregulated *MaSp1* but decreased glycine and alanine amino acids. These results highlight the inadequacy of amino acid composition as a proxy for spidroin expression.

In this study, we directly measured the expression levels of three spidroins in cobweb weaving spiders fed different diets. Our targeted genes encode proteins critical to prey capture: MaSp1 and MaSp2, the primary components of the cobweb, and MiSp, which is used to wrap prey by cobweb weavers [4]. Cobwebs are more likely to catch walking prey, such as crickets or caterpillars, than are orb-webs, which are designed to catch flying prey. However, flying insects can also be found in cobwebs, and cobweb weavers are generalist, opportunistic feeders. In our experiment, we first acclimated common house spiders, *Parasteatoda tepidariorum* (Theridiidae), to the lab, and then split them into three groups: immediate dissection, cricket-fed for one month, or fly-fed for one month. Following the feeding trial, the major and minor ampullate glands were separately dissected. We used quantitative polymerase chain reaction (qPCR) to determine mRNA levels from the major ampullate and minor ampullate glands for *MaSp1*, *MaSp2*, and *MiSp*. We hypothesized that spiders fed the high-energy prey crickets would have higher *MaSp1:MaSp2* ratio than fly-fed spiders, because increased *MaSp1* should increase the strength of cobweb fibers [7]. In addition, we predicted higher expression of *MiSp* in cricket-fed than fly-fed spiders, because house spiders would need more *MiSp* to wrap the larger cricket relative to the smaller flies.

Our work represents one of a very few studies to directly measure gene expression of *MaSp1* and *MaSp2* in response to environmental conditions, and the first to measure *MiSp* response. Because we separately analyzed major and minor ampullate glands we were also able to compare spidroin ratios and absolute abundance between the gland types and in response to diet. Instead of finding a shift in relative expression of spidroins, we found that our spiders maintained relatively high expression levels of all three spidroins under all conditions. In fact, we found that spiders may even shift resources toward spidroin production and away from expression of other genes when faced with a low-quality diet. Our results highlight the importance of silk gene expression in the life history of spiders.

## Materials and methods

We collected 61 adult or penultimate female *P. tepidariorum* in June 2014 in Lexington, Virginia, U.S.A. (Rockbridge County) outside of buildings and in the parking garage of Washington and Lee University campus. All individuals were mature by the time of dissection. The spiders were split into three groups, with each group having the same average mass (Fig 1A), by ranking the spiders by mass, and then assigning individuals to the three groups in the order they were ranked. We individually housed the spiders in cylindrical plastic containers with a paper frame to support the webs (11.5cm across the lid, 9cm across the bottom, and 7.5cm in height). All spiders were fed one larval mealworm, *Tenebrio molitar*, (~100 mg/mealworm) every two days for a total of four mealworms. The day after the fourth mealworm was fed, all webs were destroyed to stimulate silk gene expression.

Spiders in the first group were dissected three days after web destruction (hereafter, baseline). We fed the second group of spiders one cricket (~35 mg/cricket) per week and the third group five flies (~7 mg/fly) per week (1–2 times per week) to ensure equal biomass of prey between the two groups (as in [20, 22]). At two and four weeks into the feeding trial, we destroyed the webs of the spiders. We then weighed and dissected the spiders two days after the last web destruction. Each spider was subject to $CO_2$ exposure for two minutes and then dissected under 0.15 M sodium chloride, 0.015 M sodium citrate buffer. Following removal, the major and minor ampullate glands were separately frozen in liquid nitrogen and stored at -80˚ C.

Total RNA was extracted from individual spider's major ampullate glands, and separately from the minor ampullate glands, using the RNeasy-Micro kit (QIAGEN), which includes

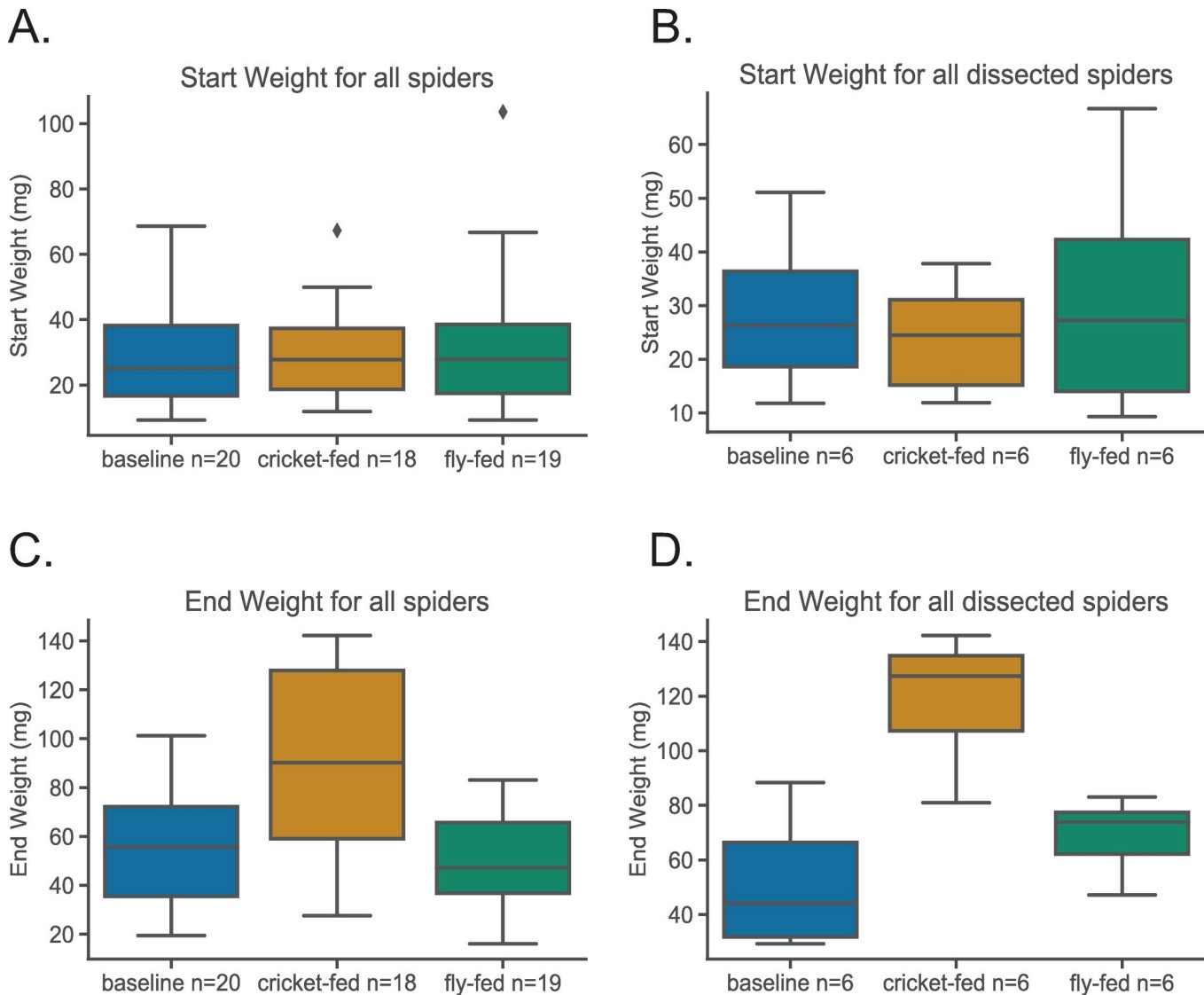

**Fig 1. Weights of spiders.** Within the box plots, heavy black lines indicate medians and box hinges are interquartile ranges. Whiskers extend to the largest and smallest values up to 1.5 times the interquartile range. Dots indicate outliers beyond 1.5 times the interquartile range. (A) all spiders start weight: ANOVA results: F = 0.1202, p = 0.89. (B) dissected spiders start weight: ANOVA results: F = 0.2948, p = 0.75. (C) all spiders end weight: ANOVA results: F = 10.31, p = 0.00016. Tukey test results, P (baseline-cricket) = 0.0023; P (baseline-fly) = 0.77; P (cricket-fly) = 0.00024. (D) dissected spiders end weight: ANOVA results: F = 16.85, p = 0.00015. Tukey test results, P (baseline-cricket) = 0.001; P (baseline-fly) = 0.33; P (cricket-fly) = 0.00024.

DNase treatment. RNA integrity was assessed by denaturing with formamide, electrophoresing on 1% agarose gels, and staining with SYBR™ Gold (Invitrogen). RNA concentration was measured with a NanoDrop 1000 (Thermo Scientific). We synthesized cDNA from 100ng of total RNA with Superscript III® (Invitrogen) primed from an anchored oligo(dT) primer as described in [26].

We amplified the C-terminal encoding regions of *P. tepidariorum MaSp1*, *MaSp2*, *MiSp*, and *Actin* using primers designed from sequences of cDNA clones (library generated by [18]; Table 1). Major ampullate and minor ampullate cDNAs were used as templates for *MaSp1* and *MaSp2* amplification. Initial experiments indicated no *MiSp* expression in major ampullate glands (Cq values equivalent to no template controls). Thereafter, only minor ampullate

**Table 1. Primers and parameters for quantitative PCR.**

| Gene | Accession Number | Forward Primer (5' to 3') | Reverse Primer (5' to 3') | Annealing Temperature | Efficiency | product length |
|------|------------------|---------------------------|---------------------------|------------------------|------------|----------------|
| *MaSp1* | MH367500 | AACCCTGGAGCCTCTGACTG | GCGCCATAGTTGATGTTTCC | 60.4˚C | 106% | 109 bp |
| *MaSp2* | MH367501 | SGTTAGCTTCTGGAGGACCAGTT | GAAGCACCAGGATTGGATGA | 60.4˚C | 96% | 93 bp |
| *MiSp* | KX584022 | CTCTGGAGCATTTCAATCCAG | AACCGAGAACAGCTCCTAAAG | 60.6˚C | 95% | 283 bp |
| *Actin* | JZ530978 | ACGAACGATTCCGTTGTCC | AATACCGCAGGACTCCATACC | 60.2˚C | 98% | 147 bp |

cDNA was used for *MiSp* amplification. qPCR amplification was performed using the MylQ5 thermocycler (BioRad) and associated software (Version 2.0). Each 20 microliter reaction volume contained the equivalent of 2.0 ng RNA of the template cDNA and 200 nanomoles of each primer in 1 X SYBR Green Supermix (BioRad). The cycling conditions included one step at 95˚C for 3 min, followed by 40 cycles of 15 sec denaturation at 94˚C, 30 sec annealing (Table 1), and 30 sec extension at 72˚ C. Each reaction concluded with a 65–95˚C melt curve analysis of 0.5˚C increments every 5 sec to ensure single product amplification. Each biological sample was amplified at least 3 times (3–9 technical replicates, S1 File).

We included standard curves with every qPCR reaction. A standard curve was made from serial dilutions of cDNA clones for *MaSp1*, *MaSp2*, *MiSp*, and *Actin* (Table 1). Transcript abundance for each biological sample was calculated by inputting the mean Cq of technical replicates into the best-fit line of regression for the appropriate standard curve (S2 File). We determined efficiency of each reaction using the equation Efficiency = $-1+10^{(-1/\text{slope of standard curve})}$ (Agilent Genomics). We additionally calculated ratios of transcript abundance to determine expression levels of one gene relative to another gene and to account for any differences in underlying mRNA levels among individuals. We did not calculate relative gene expression levels as ΔCt because primer efficiencies varied among genes (Table 1). Neither did we use ΔΔCt because we could not detect all our silk genes in both gland types and did not have another reference tissue. We tested for significant differences among feeding groups using ANOVA followed by post-hoc pairwise testing implemented in R or Python.

## Results

At the end of the feeding trial, cricket-fed spiders weighed significantly more than baseline and fly-fed spiders (Fig 1 and S1 Table). For all spidroin genes, absolute transcript abundance tended to be lower in cricket and fly-fed spiders relative to baseline for both major and minor ampullate glands. This pattern was only significant ($p<0.05$) for *MiSp* transcript abundance in minor ampullate glands with fly-fed spiders significantly lower than baseline spiders (Fig 2B; Table 2). *MaSp2* in major ampullate glands approached significance ($p = 0.056$), with fly and cricket-fed spiders lower than baseline (Fig 2A). Although *Actin* transcript abundance did not significantly differ among feeding groups (Table 2), fly-fed spiders have lower transcript abundance than other feeding groups in minor ampullate glands (Fig 2C) and both fly-fed and cricket-fed had lower transcript abundance than baseline in major ampullate glands (Fig 2C). Thus, when each spidroin gene was measured relative to actin, the pattern of expression changed. This was especially true for *MiSp* in minor ampullate glands. Although not quite significant ($p = 0.06$), cricket-fed spiders had the lowest *MiSp* relative to actin (Fig 2D). This same pattern of cricket-fed spiders having the lowest spidroin expression relative to actin in minor ampullate glands held true for *MaSp1* and *MaSp2* (Fig 3D and 3F). In major ampullate glands, fly-fed spiders had higher average *MaSp1*:actin and *MaSp2*:actin expression than the other two feeding groups (Fig 3C and 3E).

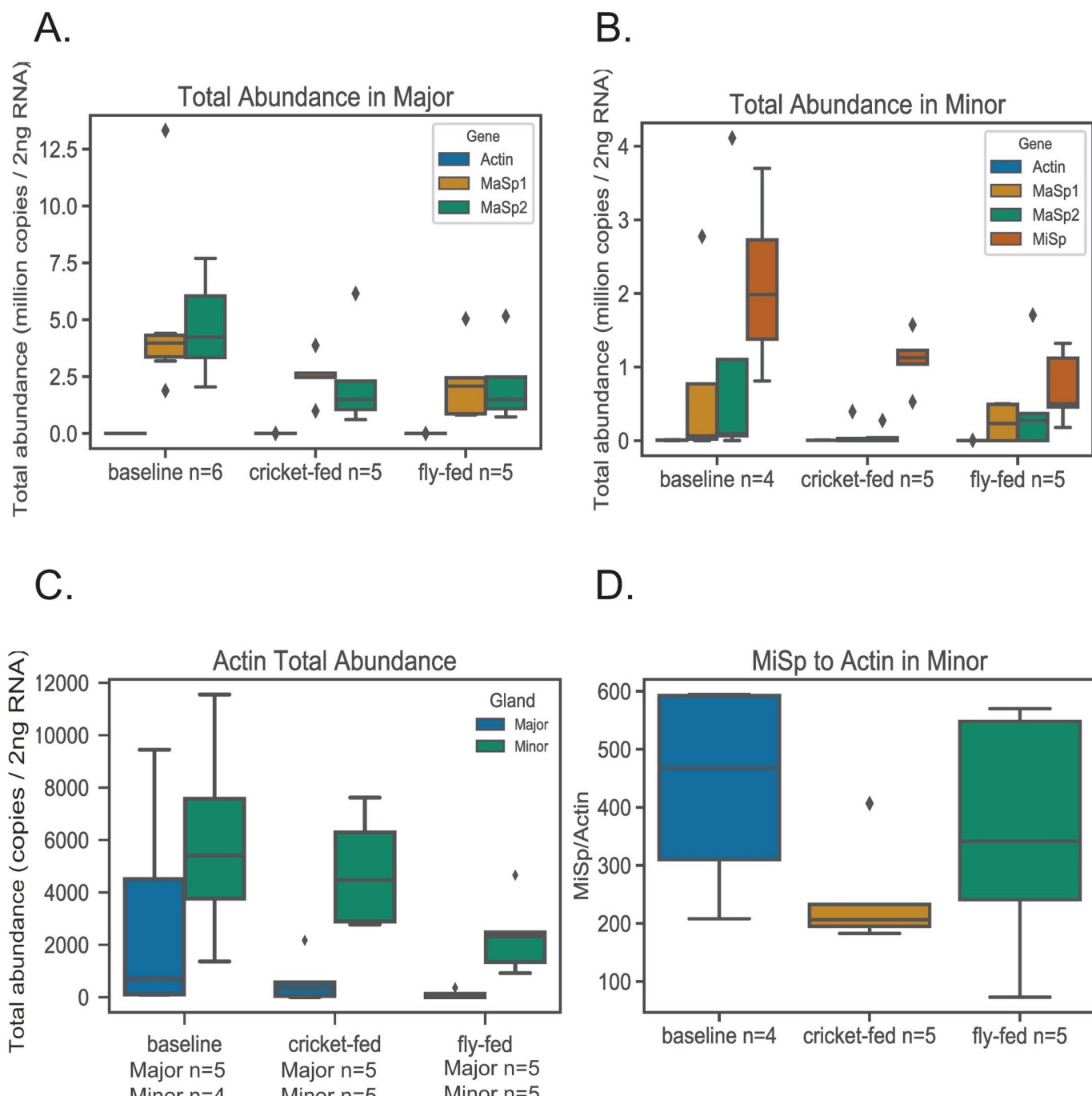

**Fig 2. Transcript abundance in major and minor ampullate glands.** Medians and ranges for box plots as in Fig 1. (A) Total abundance of *MaSp1*, *MaSp2*, and *actin* in major ampullate glands. (B) Total abundance of *MaSp1*, *MaSp2*, *MiSp*, and *Actin* in minor ampullate glands. (C) *Actin* total abundance in major and minor ampullate glands. (D) *MiSp* abundance / *Actin* abundance in minor ampullate glands.

The ratio of MaSp1:MaSp2 is modelled as an important determinant of dragline silk material properties [15, 27]. We found the transcript ratio for *MaSp1*:*MaSp2* was very similar between major ampullate (mean = 1.60) and minor ampullate (mean = 1.61) glands (Fig 3A and 3B). Although some individuals deviated considerably from the mean (S2 File), the ratio

**Table 2. Results of one-way ANOVA comparing absolute or ratios of transcript abundance among three feeding groups.**

| Gland type | Response variable | p-value | F-value |
|---|---|---|---|
| Major | Actin | 0.22 | 1.77 |
| | MaSp1 | 0.20 | 1.84 |
| | MaSp2 | 0.056 | 3.63 |
| | MaSp1:MaSp2 | 0.064 | 3.41 |
| | MaSp1:Actin | 0.18 | 2.05 |
| | MaSp2:Actin | 0.20 | 1.89 |
| Minor | Actin | 0.17 | 2.11 |
| | MaSp1 | 0.44 | 0.89 |
| | MaSp2 | 0.45 | 0.87 |
| | MaSp1:MaSp2 | 0.34 | 1.21 |
| | MaSp1:Actin | 0.35 | 1.17 |
| | MaSp2:Actin | 0.33 | 1.21 |
| | MiSp | 0.046 | 4.12 |
| | MiSp:MaSp1 | 0.32 | 1.27 |
| | MiSp:MaSp2 | 0.38 | 1.07 |
| | MiSp:Actin | 0.06 | 3.63 |

of *MaSp1*:*MaSp2* did not significantly differ among feeding groups for major ampullate glands or minor ampullate glands (Table 2), and there was no interaction between gland type and diet (two-way ANOVA: Overall model $F_{(5, 24)} = 1.371$, $P = 0.2702$).

## Discussion

We found little difference in the absolute or relative abundance of dragline gene transcripts, *MaSp1* and *MaSp2* among diet groups. This finding contradicts our expectation that house spiders fed high energy crickets would have higher *MaSp1*:*MaSp2* expression ratio compared to those fed low-energy flies. Also, we had originally predicted an increased level of *MiSp* expression in cricket-fed spiders compared to the fly-fed spiders. We found the fly-fed group to have the lowest absolute *MiSp* abundance but when measured relative to *Actin*, cricket fed spiders had the lowest *MiSp* expression. Below we discuss why cobweb weavers might differ from orb-web weavers in their spidroin expression response to diet and how major ampullate silk material properties could change without concomitant gene expression changes. We further discuss our unexpected *MiSp* findings, as the first measure of expression of this gene in response to environmental conditions, and the likely role of starvation in explaining our results.

Based on similar feeding trials with orb-web weavers, we expected prey type to affect major ampullate spidroin gene expression [20, 22, 23]. Specifically, we expected the cricket-fed spiders to have increased expression of *MaSp1*, which should increase fiber strength [7]; house spiders fed crickets have stronger major ampullate silk than house spiders fed low-energy pillbugs [28]. However, our results suggest that neither of the major ampullate spidroins significantly respond to prey type. Our results may have differed from those found for orb-web weavers, because orb-web weavers rebuild their web every day, unlike cobweb weavers, which add to their existing web. It is likely that maintaining a cobweb is less energy consuming than rebuilding an orb-web daily, resulting in no significant change in major ampullate spidroin expression for cobweb weavers eating different prey.

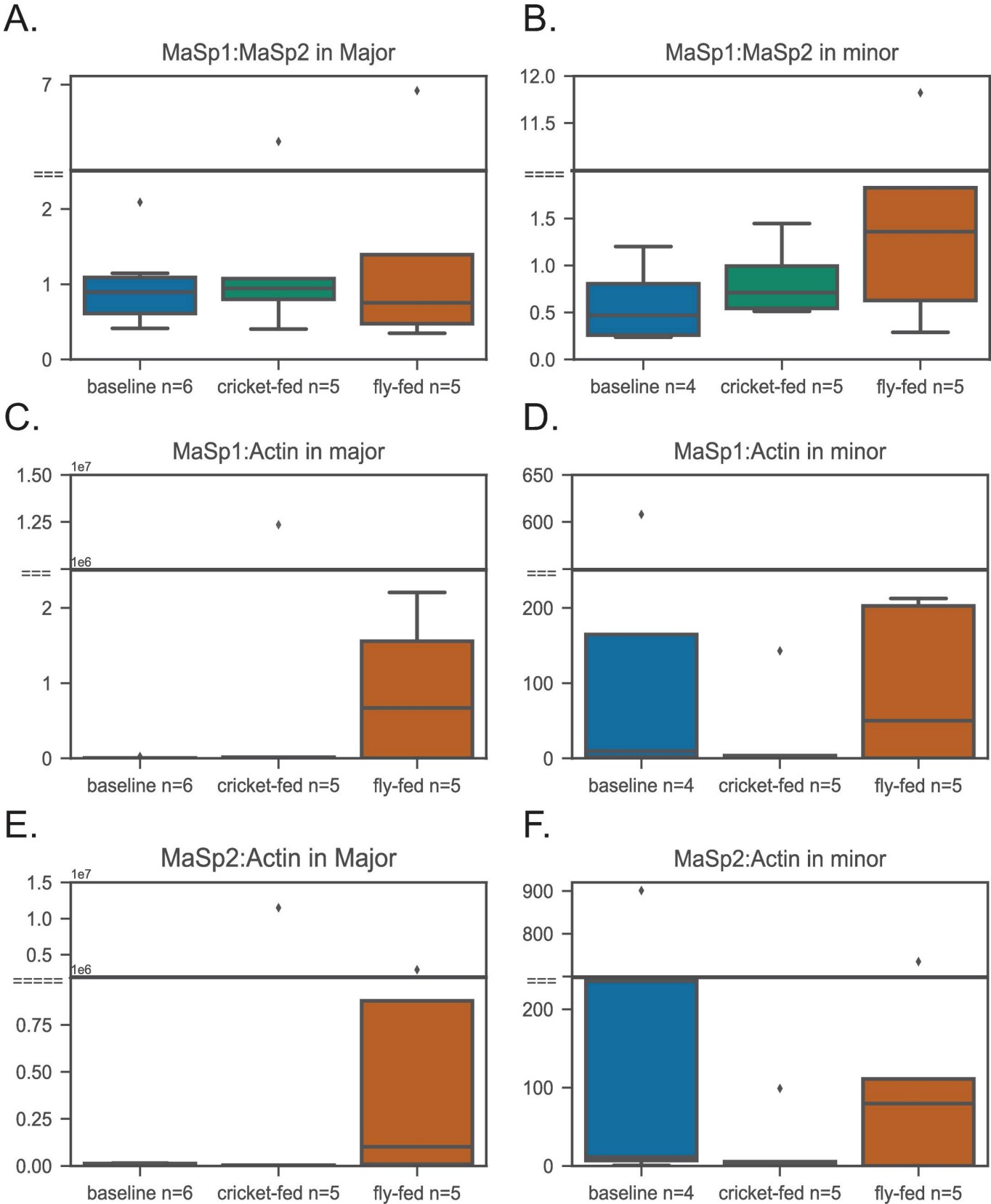

**Fig 3. *MaSp1* and *MaSp2* abundance relative to each other and to *Actin*.** Medians and ranges for box plots as in Fig 1. (A) *MaSp1/MaSp2* in major ampullate glands. (B) *MaSp1/MaSp2* in minor ampullate glands. (C) *MaSp1/Actin* ratio in major ampullate glands. (D) *MaSp1/Actin* in minor ampullate glands. (E) *MaSp2/Actin* in major ampullate glands (F) *MaSp2/Actin* in minor ampullate glands.

Another functional difference between cobwebs and orb-webs is that cobwebs incorporate major ampullate silk as structural support for the entire three-dimensional webs, whereas orb-webs use major ampullate silk to absorb the impact of flying prey [25, 29]. Thus, it is possible the major ampullate silk of orb-web weavers has greater inherent variability to adjust the web functionality, which may be triggered by changes in nutrient uptake [25, 28]. It was found under nutrient deprivation, the major ampullate silk of a cobweb species, *Latrodectus hasselti*, was unaffected in terms of material properties and amino acid composition, although expression of some MaSp-encoding genes increased and others decreased relative to a reference gene [25]. Thus, major ampullate silk of cobweb weavers may not have as much inherent variability as orb-web weavers, and are thus less sensitive to changes in nutrient uptake [25]. Our findings could support this hypothesis since we saw minimal changes in major ampullate spidroins across feeding groups, suggesting major ampullate silk of *P. tepidariorum*, was not sensitive to the changes in prey-type. Nonetheless, our results show a fair bit of individual variation in terms of spidroin expression in both major and minor ampullate glands in each feeding group (S2 File). It is likely that major ampullate spidroin expression in house spiders is variable but is not strongly influenced by changes in nutrient uptake, at least as adults.

Although the major ampullate silk of the cobweb weaver *L. hasselti* [25] had minimal changes in material properties in response to nutrient deprivation, the major ampullate silk of house spiders did have changes in material properties in response to prey type. In a study that investigated the silk mechanical properties of *P. tepidariorum* fed either pillbugs or crickets, those fed pillbugs had an average gumfoot thread strength of 1200 MPa and extensibility of about 0.4 ln(mm/mm). The cricket-fed spiders' draglines had overall higher strength (1550 MPa) compared to the pillbug-fed spiders as well as modest increases in extensibility (0.43 ln (mm/mm)) [28]. Although we did not measure material properties, it is possible the silk strength differed in response to the changes in nutrient intake among the individuals despite an insignificant difference in *MaSp1:MaSp2* ratio among the feeding groups for major and minor ampullate glands.

Comparable to our finding that cricket-fed spiders weighed significantly more than the fly-fed spiders, Boutry and Blackledge [28] found cricket-fed spiders weighed almost twice as much as the pill-bug fed spiders. Boutry and Blackledge [28] suggested spiders extract greater biomass from crickets, thus making the difference in material and mechanical properties of silk a response to different levels of starvation. A decrease in expression of *MaSp1* relative to *MaSp2* when starved (fed pillbugs) could be one mechanism through which these house spiders changed the strength of their spider silk. However, a more likely possibility, given our findings of no change in *MaSp1:MaSp2* in house spiders, is that spider body mass plays a direct role in variation of material properties of silk; Boutry and Blackledge [28] found thread diameter and failure load correlated to body mass. Our study aligns with these findings as we found no difference in spidroin expression based on diet, suggesting spider body mass may play a more significant role in modulating silk properties in which spiders may tune their silk in response to their own body mass.

Our spiders in their original environment had access to both low- and high-quality diet, so it is unknown if the spiders were accustomed to a suboptimal diet before capture. In the lab, the spiders were standardized at the beginning of the experiment by being fed larval mealworms, *Tenebrio molitor*, as the baseline diet. A larval mealworm has 187 g/kg of protein and

134 g/kg of fat [30]. The mealworms are thus highly nutritious. Once on the experimental diet, the spiders fed crickets likely continued to receive an optimal diet, whereas the fly-fed spiders received a suboptimal diet. An adult cricket has 205 g/kg of protein while a fly has 197 g/kg of protein [30, 31]. A cricket has 68 g/kg of fat and 1,402 kcal/kg of calories, whereas a fly has only 19 g/kg of fat and 918 kcal/kg of calories. Thus, the group that was fed crickets received greater fat and calories even though the spiders were fed equal biomass of crickets and flies. Thus, the spiders fed flies received less nutrient intake, making the fly diet suboptimal compared to the cricket diet. We found the spiders on this suboptimal diet to maintain dragline gene expression as there were no significant differences in MaSp1 or MaSp2 expression compared to the cricket-fed group.

Although we did not observe any effect of diet on dragline genes, we did find evidence for modulated gene expression in the minor ampullate glands. There is no previous literature on *MiSp* expression based on diet for any spider species. We had predicted an increased level of *MiSp* expression in our spiders fed high energy crickets because cobweb weavers use minor ampullate silk for prey wrapping, whereas orb-web weavers use them for the auxiliary spiral of their web. Crickets, due to their larger size, struggle more in the web and we observed house spiders throwing more silk on crickets than flies. We found slightly higher absolute abundance of *MiSp* in the cricket-fed group, but the fly-fed group was significantly lower than the baseline group in which it is possible even more silk is thrown on mealworms. Interestingly, when normalized against *Actin* transcript abundance, cricket-fed spiders had the lowest *MiSp* abundance, which was borderline significant (Table 1). Previous research has suggested there is a metabolic cost to synthesizing amino acids used in silk production [32], thus, predicting decreased expression of costly spidroins. Due to the metabolic costs of silk synthesis, the starving fly-fed spiders may have lowered the expression of other genes in order to maintain sufficient *MiSp* levels. Our results suggest the fly-fed group had lower overall gene expression, but potentially upregulate their expression of *MiSp* relative to other transcripts, like *Actin*, in the face of a low-quality diet in order to maintain sufficient prey-wrapping silk.

While most previous studies use amino acid composition to study spidroin levels, our study used qPCR as a direct measure of gene expression levels. A further strength of our approach was the ability to measure absolute abundance of spidroin transcripts by reference to a dilution series of cDNA clones containing the genes of interest (S1 File). A possible limitation of our qPCR design was the choice of the reference gene. *Actin* was chosen as a reference gene because it does not differ much among tissues [33] and has been used in multiple studies of tissue-specific spidroin expression (e.g. [4, 34–36]). However, based on our results, *Actin* may respond to diet as it was lower, albeit not significantly different, in the cricket and fly-fed spiders than the baseline spiders. Therefore, *Actin* might not be the best reference gene to normalize spidroin expression. Other reference genes that could have been used include *g3dph* [25], and calreticulin [26]. Ideally, multiple reference genes should be used to confirm our hypothesis that house spiders decrease housekeeping gene expression in order to maintain spidroin expression.

Our study went beyond others in directly measuring expression levels of major ampullate and minor ampullate encoding silk genes, but there are multiple additional silk types that contribute to prey capture in cobweb and orb-web weaving spiders. For instance, aciniform silk and aggregate gluey silk are used to wrap prey by cobweb weavers [34, 37]. Cobweb weavers also place aggregate glue on gumfoot lines, major ampullate threads that radiate from the web to the ground [3, 37]. These gumfoot lines are especially adept at catching walking prey. To fully understand the impact of diet on prey capture silks, the expression of the aciniform and aggregate spidroin encoding genes should be measured in addition to major and minor ampullate spidroin encoding genes. In order to tie gene expression levels to protein

composition, quantitative mass spectrometry of cobwebs, gumfoot lines, and prey wrapping material should be performed, or minimally the amino acid composition of each of the materials should be measured. With the advent of transcriptomes, genomes, and thus full-length transcripts of multiple spidroins (e.g. [12, 38–42]), these types of direct comparisons will become increasingly feasible.

In conclusion, the absolute transcript abundance for all spidroins tended to be lower in cricket and fly-fed spiders compared to baseline for both major and minor ampullate glands. However, once compared against *Actin*, it was found cricket-fed spiders had the lowest *MiSp* expression. Additionally, in major ampullate glands, the fly-fed spiders had higher average *MaSp1*:*Actin* and *MaSp2*:*Actin* expression than the other two feeding groups. The low weights of fly-fed spiders suggest these spiders were likely starved during the feeding regime. Despite this sub-optimal diet, these fly-fed spiders maintained expression of three spidroin genes comparable to the cricket-fed or baseline spiders. Therefore, house spiders appear to maintain silk gene expression, or possibly even increase silk gene expression relative to other genes, even in the face of a low-quality diet.

## Supporting information

**S1 File. Raw Cq values from qPCR.** Includes separate sheets for each gene in each gland type including raw Cq for all technical replicates and calculations for estimating absolute abundance.
(XLSX)

**S2 File.**
(DOCX)

**S1 Table. The weights of spiders before and after experiment with notes about egg production.**
(XLSX)

## Acknowledgments

We thank Robert Haney, Evelyn Schwager, and Sean Blamires for comments on drafts of this manuscript. We thank Fred LaRiviere for advice on qPCR and access to his thermal cycler.

## Author Contributions

**Conceptualization:** Jeremy Miller, Nadia A. Ayoub.

**Data curation:** Jeremy Miller, Nadia A. Ayoub.

**Formal analysis:** Jeremy Miller, Jannelle Vienneau-Hathaway, Enkhbileg Dendev, Merrina Lan, Nadia A. Ayoub.

**Funding acquisition:** Nadia A. Ayoub.

**Investigation:** Jannelle Vienneau-Hathaway, Nadia A. Ayoub.

**Methodology:** Jeremy Miller, Jannelle Vienneau-Hathaway, Nadia A. Ayoub.

**Project administration:** Nadia A. Ayoub.

**Resources:** Nadia A. Ayoub.

**Supervision:** Nadia A. Ayoub.

**Visualization:** Merrina Lan.

**Writing – original draft:** Jeremy Miller, Jannelle Vienneau-Hathaway, Enkhbileg Dendev, Nadia A. Ayoub.

**Writing – review & editing:** Jeremy Miller, Jannelle Vienneau-Hathaway, Enkhbileg Dendev, Merrina Lan, Nadia A. Ayoub.

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
