## [Decision Letter · Decision Letter 0]

8 Sep 2020

PONE-D-20-22090

The common house spider, *Parasteatoda tepidariorum*, maintains silk gene expression on sub-optimal diet

PLOS ONE

Dear Dr. Ayoub,

Thank you for submitting your manuscript to PLOS ONE. After careful consideration, we feel that it has merit but does not fully meet PLOS ONE’s publication criteria as it currently stands. Therefore, we invite you to submit a revised version of the manuscript that addresses the points raised during the review process.

We look forward to receiving your revised manuscript.

Kind regards,

Giovanni Signore

Academic Editor

PLOS ONE

Journal Requirements:

Reviewers' comments:

Reviewer's Responses to Questions

**Comments to the Author**

1. Is the manuscript technically sound, and do the data support the conclusions?

Reviewer #1: Yes

Reviewer #2: Yes

2. Has the statistical analysis been performed appropriately and rigorously? 

Reviewer #1: Yes

Reviewer #2: Yes

3. Have the authors made all data underlying the findings in their manuscript fully available?

Reviewer #1: Yes

Reviewer #2: Yes

4. Is the manuscript presented in an intelligible fashion and written in standard English?

Reviewer #1: Yes

Reviewer #2: Yes

5. Review Comments to the Author

Reviewer #1: In this study, the authors investigated the extent to which the characteristics of P. tepidariorum silk can be influenced by different types of diet, based on the transcript of 3 spidroin genes. The strong point of this manuscript is using qPCR, but it laks clarity and focus in some areas that can be improved. Furthermore, discussion needs to be better documented. There is no comment for a major revision.

Specific comments and/or areas for improvement are:

-Abstract

Line 13: "Orb-web weaving spiders and their relatives...". Perhaps authors shuold start by talking about P. tepidariourm or cobweb weaving spiders.

-Introduction

It should highlight the methodological challenges of previous studies and the void that this study can fill. As currently written, it is difficult to see how this study will add to the literature / resolve the controversy. The importance of measuring levels of gene expression may be more stressed.

-Material and Methods

I would like the authors to provide more information on choice of mealworms as baseline diet, mg/mealworms and their nutritional values.

-Results

No comment.

-Discussion

This section would also benefit from better organization of the findings. The authors are speculating that P. tepidariorum spiders body mass is a proxy for material properties of silk. Maybe it is necessary to report in the main text (Results) significant weight differences between crickets-fed spiders and flies-fed spiders.

Reviewer #2: The manuscript ‘The common house spider Parasteatoda tepidoiorum maintains…’ was a very interesting, well organized, original and timely manuscript that I strongly advise be published in Plos One.

While the experimental design, analysis and interpretation is generally very good. I do have issues that I will expect the authors to nonetheless address. These are:

Firstly, the paper compares the expression of 3 spidroin genes in the subject spider across diets; (i) MaSp1, (ii) MaSp2, and (3) MiSp. It is reasonable for the purpose of the study to focus in on these. However, others may be involved (other MaSps, AcSp etc..), especially when it comes to affecting the amino acid compositions of the silks produced by the spiders. As the major point to doing the study is to show why it is that you may or may not get variation in silk amino acid compositions without necessarily concomitant variations in gene expression, any additional genes potentially playing a part should be mentioned. The authors should particularly cite the 3 silk full transcript studies done to date- on (i) Nephila clavipes by Babb et al in Nature Genetics, (ii) on Araneus ventricosus by Kono et al. in Scientific Reports, and on Caerostris darwinii by Garb et al in Communications Biol.

On line 46-48 the authors mention the trend of proline influencing silk extensibility across spider species. There is a new paper that shows this trend much better (Craig HC et al. 2020. Journal of the Royal Society Interface, in press who built a 85 species phylogeny) than the paper cited. Please incorporate this citation. The abovementioned Craig paper also suggests that it may be when proline gets hydroxylated post-expression that it has is effect on extensibility. Please incorporate discussion here on the additional effects of hydroxylated proline on silk extensibility.

The point made about not using the Delta CT method is well presented and I can concede the authors made the right decision. I would like them to make comments on how they might back compare their results with studies that did use the Delta CT method. Also the point about the lack of a good reference gene (therein and later in the Discussion) is valid, but I wonder if alternatives had been considered. This is especially pertinent in light of the few studies running these kinds of expression analyses having similar trouble finding good reference genes. In the analyses of Blamires et al 2018, for instance, the g3dph reference gene varied across samples, but the authors cross-compared it with other genes expressed to make conservative estimates of their delta-delta CT values.

At some stage in the Introduction describe the spiders ‘normal’ diet and how they feed, as if the diets fed here are unusual in some way, or the insects are difficult to handle/wrap up/consume, or require behavioural changes to do so, this itself might affect the expression of particular spidroin genes.

At line 223 cite Boutry & Blackledge (ref 26).

Lastly, I’d like to see more in the Discussion about some proposed mechanisms by which diet might induce variability in spidroin gene expression in spiders. Metabolic costs of synthesizing the amino acids have been hypothesized (see C. Craig’s 2003 book and a 1999 Int J. Biol. Macromol paper) as a mechanism. There might be something in this in light of similar differentiated expression happening when spiders exposed to insecticides (Benamu et al. 2017. Chemosphere). What else might be feasible? Are there any insights in the data of this study?

6. PLOS authors have the option to publish the peer review history of their article (what does this mean?). If published, this will include your full peer review and any attached files.

Reviewer #1: No

Reviewer #2: **Yes: **Sean Blamires

---

## [Author Response · Author response to Decision Letter 0]

26 Oct 2020

Response to Reviewers’ Comments have been uploaded as a Microsoft Word document and are also pasted below.

Reviewers’ comments are pasted below in black text. Our responses are below each comment in bolded text.

Review Comments to the Author: 

Reviewer #1: 

In this study, the authors investigated the extent to which the characteristics of P. tepidariorum silk can be influenced by different types of diet, based on the transcript of 3 spidroin genes. The strong point of this manuscript is using qPCR, but it laks clarity and focus in some areas that can be improved. Furthermore, discussion needs to be better documented. There is no comment for a major revision.

**Thank you for the positive feedback and suggestions for clarification. Our efforts to clarify the strengths and outcomes of our work are detailed below.**

Specific comments and/or areas for improvement are:

-Abstract

Line 13: "Orb-web weaving spiders and their relatives...". Perhaps authors shuold start by talking about P. tepidariourm or cobweb weaving spiders.

**We changed the first sentence of the abstract to “Cobweb weaving spiders and their relatives…” and we added “Orb-web and cobweb weaving spiders…” to the second sentence of the introduction to clarify that both types of spiders are members of the diverse Araneoidea. We further clarified aspects of web building and feeding behaviors unique to cobweb weavers in the second to last paragraph of the Introduction.**

-Introduction

It should highlight the methodological challenges of previous studies and the void that this study can fill. As currently written, it is difficult to see how this study will add to the literature / resolve the controversy. The importance of measuring levels of gene expression may be more stressed.

**We added a paragraph to the end of the introduction to better highlight the strengths of our study, which include using qPCR to directly measure gene expression levels, measuring 3 spidroin genes important for prey capture – with our study being the first to explore minor ampullate encoding genes, and analyzing major and minor ampullate glands separately. Additional strengths that we clarified in the Discussion section include the fact that our qPCR primers were designed from known sequences, and that we had cDNA clones of these sequences that could be used as reference to generate absolute abundance measurements (lines 288-291, 299-300).**

-Material and Methods

I would like the authors to provide more information on choice of mealworms as baseline diet, mg/mealworms and their nutritional values.

**In the methods (lines 113-114), we added the species of mealworms, Tenebrio molitar, and the average mass of a mealworm. In the discussion, we added a comparison of the nutritional content of mealworms to the experimental diets of crickets and flies. The mealworms and crickets were highly nutritious diets for the spiders, while the flies likely represented a suboptimal diet (lines 263-267). **

-Results

No comment.

-Discussion

This section would also benefit from better organization of the findings. The authors are speculating that P. tepidariorum spiders body mass is a proxy for material properties of silk. Maybe it is necessary to report in the main text (Results) significant weight differences between crickets-fed spiders and flies-fed spiders.

**We reported the weights of spiders at the beginning and end of the experiment in Figure 1, which is first referenced in the Methods. We additionally reported the significant weight differences after the feeding trials in the first sentence of the results (lines 166-167). We also added an outline of the Discussion to the first paragraph to help guide the reader, as well as added text throughout the discussion to clarify the strengths of the study.**

Reviewer #2: The manuscript ‘The common house spider Parasteatoda tepidoiorum maintains…’ was a very interesting, well organized, original and timely manuscript that I strongly advise be published in Plos One.

While the experimental design, analysis and interpretation is generally very good. I do have issues that I will expect the authors to nonetheless address. These are:

Firstly, the paper compares the expression of 3 spidroin genes in the subject spider across diets; (i) MaSp1, (ii) MaSp2, and (3) MiSp. It is reasonable for the purpose of the study to focus in on these. However, others may be involved (other MaSps, AcSp etc..), especially when it comes to affecting the amino acid compositions of the silks produced by the spiders. As the major point to doing the study is to show why it is that you may or may not get variation in silk amino acid compositions without necessarily concomitant variations in gene expression, any additional genes potentially playing a part should be mentioned. The authors should particularly cite the 3 silk full transcript studies done to date- on (i) Nephila clavipes by Babb et al in Nature Genetics, (ii) on Araneus ventricosus by Kono et al. in Scientific Reports, and on Caerostris darwinii by Garb et al in Communications Biol.

**Thank you for the positive feedback and the many helpful suggestions. There are certainly other silk proteins involved in prey capture by cobweb weavers, such as aciniform silk and aggregate gluey silk. Exploring these genes was beyond the scope of the study we could conduct at the time but agree that in the future, the expression of aciniform and aggregate spidroin encoding genes should be measured in addition to the genes we analyzed. As the reviewer suggests, as full-length spidroin encoding genes become more commonly available, it will be reasonable to measure gene expression and amino acid content or, even better, quantify proteins with mass spectrometry-based proteomics. We added these ideas in the 2nd to last paragraph of the discussion, as well as the suggested references and additional references to full-length spidroin transcripts.**

On line 46-48 the authors mention the trend of proline influencing silk extensibility across spider species. There is a new paper that shows this trend much better (Craig HC et al. 2020. Journal of the Royal Society Interface, in press who built a 85 species phylogeny) than the paper cited. Please incorporate this citation. The abovementioned Craig paper also suggests that it may be when proline gets hydroxylated post-expression that it has is effect on extensibility. Please incorporate discussion here on the additional effects of hydroxylated proline on silk extensibility.

**Thank you for bringing this publication to our attention! We could not actually access it when we first received the review, but it was published and available a couple of weeks later. We agree that Craig et al. (2020) confirms and expands on the trend previously demonstrated with only a handful of species. We added this reference and a description of how proline contributes to extensibility of silks to the second paragraph of the introduction (lines 47-51).**

The point made about not using the Delta CT method is well presented and I can concede the authors made the right decision. I would like them to make comments on how they might back compare their results with studies that did use the Delta CT method. Also the point about the lack of a good reference gene (therein and later in the Discussion) is valid, but I wonder if alternatives had been considered. This is especially pertinent in light of the few studies running these kinds of expression analyses having similar trouble finding good reference genes. In the analyses of Blamires et al 2018, for instance, the g3dph reference gene varied across samples, but the authors cross-compared it with other genes expressed to make conservative estimates of their delta-delta CT values.

**In the third to last paragraph, we added that a further strength of our approach was the ability to measure absolute abundance of spidroin transcripts by reference to a dilution series of cDNA clones containing the genes of interest. We concede that other reference genes such as g3dph and calreticulin could have been used. And state that “Ideally, multiple reference genes should be used to confirm our hypothesis that house spiders decrease housekeeping gene expression in order to maintain spidroin expression”. We also added a description of the findings for major ampullate genes in another cobweb weaver, Latrodectus hasselti to the discussion (lines 228-230).**

At some stage in the Introduction describe the spiders ‘normal’ diet and how they feed, as if the diets fed here are unusual in some way, or the insects are difficult to handle/wrap up/consume, or require behavioural changes to do so, this itself might affect the expression of particular spidroin genes.

**We added a description of how these spiders usually catch prey to the introduction (lines 86-88). Essentially, flies, crickets, and beetle/moth larvae would be typical prey items caught by house spiders in our region of Virginia. **

At line 223 cite Boutry & Blackledge (ref 26).

**Thank you for catching this missing reference. It has been added. **

Lastly, I’d like to see more in the Discussion about some proposed mechanisms by which diet might induce variability in spidroin gene expression in spiders. Metabolic costs of synthesizing the amino acids have been hypothesized (see C. Craig’s 2003 book and a 1999 Int J. Biol. Macromol paper) as a mechanism. There might be something in this in light of similar differentiated expression happening when spiders exposed to insecticides (Benamu et al. 2017. Chemosphere). What else might be feasible? Are there any insights in the data of this study?

**The metabolic cost of synthesizing expensive amino acids found in spidroins could certainly shift expression at multiple levels (e.g. transcriptional, translational). Our results cannot directly test that hypothesis, but we brought up the possibility in the discussion (lines 281-284) and speculate that spiders subjected to suboptimal environmental conditions may have to shift resources away from other proteins in order to maintain silk production, due to this metabolic cost. **

---

## [Editor Report · Decision Letter 1]

30 Oct 2020

The common house spider, *Parasteatoda tepidariorum*, maintains silk gene expression on sub-optimal diet

PONE-D-20-22090R1

Dear Dr. Ayoub,

We’re pleased to inform you that your manuscript has been judged scientifically suitable for publication and will be formally accepted for publication once it meets all outstanding technical requirements.

Kind regards,

Giovanni Signore

Academic Editor

PLOS ONE
---

## [Editor Report · Acceptance letter]

16 Nov 2020

PONE-D-20-22090R1 

The common house spider, *Parasteatoda tepidariorum*, maintains silk gene expression on sub-optimal diet 

Dear Dr. Ayoub:

I'm pleased to inform you that your manuscript has been deemed suitable for publication in PLOS ONE. Congratulations! Your manuscript is now with our production department. 

Kind regards, 

on behalf of

Dr. Giovanni Signore 

Academic Editor

PLOS ONE